# Understanding Risk Behaviors of Vietnamese Adults with Chronic Hepatitis B in an Urban Setting

**DOI:** 10.3390/ijerph16040570

**Published:** 2019-02-16

**Authors:** Thieu Van Le, Thuc Thi Minh Vu, Anh Kim Dang, Giang Thu Vu, Long Hoang Nguyen, Binh Cong Nguyen, Tung Hoang Tran, Bach Xuan Tran, Carl A. Latkin, Cyrus S.H. Ho, Roger C.M. Ho

**Affiliations:** 1Viet-Tiep Friendship Hospital, Hai Phong 180000, Vietnam; thieulv@gmail.com (T.V.L.); nguyencongbinhvt@gmail.com (B.C.N.); 2Hanoi Medical University, Hanoi 100000, Vietnam; vuminhthuc2010@yahoo.com.vn; 3Institute for Global Health Innovations, Duy Tan University, Da Nang 550000, Vietnam; kimanh.ighi@gmail.com; 4Center of Excellence in Evidence-based Medicine, Nguyen Tat Thanh University, Ho Chi Minh City 700000, Vietnam; giang.coentt@gmail.com; 5Center of Excellence in Behavioral Medicine, Nguyen Tat Thanh University, Ho Chi Minh City 700000, Vietnam; longnh.ph@gmail.com (L.H.N.); hocmroger@yahoo.com.sg (R.C.M.H.); 6Institute of Orthopaedic and Trauma Surgery, Vietnam—Germany Hospital, Hanoi 100000 Vietnam; tranhoangtung.vd@gmail.com; 7Institute for Preventive Medicine and Public Health, Hanoi Medical University, Hanoi 100000 Vietnam; 8Bloomberg School of Public Health, Johns Hopkins University, Baltimore, MD 21205, USA; carl.latkin@jhu.edu; 9Department of Psychological Medicine, National University Hospital, Singapore 119074, Singapore; cyrushosh@gmail.com; 10Department of Psychological Medicine, Yong Loo Lin School of Medicine, National University of Singapore, Singapore 119228, Singapore

**Keywords:** smoking, alcohol, sexual risk, chronic hepatitis B, Vietnam

## Abstract

Cigarette smoking and alcohol consumption can be considered as risk factors that increase the progression of chronic liver disease. Meanwhile, unprotected sex is one of the main causes of hepatitis B infection. This study aimed to explore drinking, smoking, and risky sexual behaviors among people with chronic hepatitis B virus (HBV) in a Vietnamese urban setting, as well as investigating potential associated factors. A cross-sectional study was performed in October 2018 in Viet-Tiep Hospital, Hai Phong, Vietnam. A total of 298 patients who had been diagnosed with chronic hepatitis B reported their smoking status, alcohol use, and sexual risk behavior in the last 12 months. A multivariate logistic regression model was used to identify the associated factors. It was identified that 82.5% of participants never used alcohol. The Alcohol Use Disorders Identification Test-Consumption (AUDIT-C) positive result among male patients was 7.4% (0% in female patients). In addition, 14.5% of participants were current smokers and the mean number of cigarettes per day was 7.4 (SD = 3.4). It was found that 35.4% of male patients had sex with two or more sex partners. Furthermore, 66.7% and 74.1% of participants used condoms when having sex with casual partners/one-night stands and sex workers, respectively. There was a positive correlation between monthly drinking and currently smoking. White-collar workers were less likely to have multiple sex partners within the last 12 months. Our study highlights the need for integrating counseling sessions and educational programs with treatment services.

## 1. Introduction

Hepatitis B—a liver infection caused by the hepatitis B virus (HBV)—has long been considered a threat to global health due to its high prevalence and mortality rate [1]. According to the World Health Organization, the number of people with positive HBV surface antigen (HBsAg) was as high as 257 million in 2015, while fatality records were estimated to be 887,000 in 2015 alone [2]. Hepatitis B has been particularly prevalent in Asian countries, especially in those of Southeast Asia—classified as a high endemic region with over 8% of the population being diagnosed with positive HBsAg [3].

Hepatitis B virus infection has a distinct course of progression, of which a significant step is the transition of an acute episode to a chronic condition [4]. Studies have found that the initiation of HBV chronicity is generally associated with the condition of the immune system and host factors [5,6]. Although the rate of developing chronic HBV is highest among infants (90%), adults with a competent immune system can also develop HBV chronicity at a 5–10% rate [7]. Meanwhile, chronic HBV has been considered the main cause of hepatocellular carcinoma (HCC) or primary liver cancer—the second leading cause of cancer mortality, with an annual death count of 745,000 [8]. A Global Burden of Disease Study in 2010 indicated that chronic HBV accounted for almost 45% of HCC cases investigated globally [9]. Understanding the risky behaviors of chronic HBV patients, especially in relation to HCC development, is essential to reduce HBV-related deaths and the burden of diseases. Risky behaviors are defined as behaviors that expose people to harm and negatively affect physical, economic, or psycho-social well-being [10]. Prevailing literature has identified risky sexual behavior as one of the main risk factors for the transmission of HBV [11,12], while recent research has reported associations of alcohol consumption [13] and cigarette smoking with a risk of liver cancer in those infected with chronic HBV [14]. 

The prevalence of chronic HBV in Vietnam is substantial—HbsAg positive infection was found to be in the range of 9 to 14% of the population in the two largest cities of Vietnam [15], with a projection of total chronic HBV cases reaching 8 million by 2025 [16]. However, there is little literature that contributes to identifying the risk behaviors of Vietnamese chronic HBV patients. Thus, this study aimed to explore these risk behaviors (in particular drinking, smoking, and risky sexual behavior) among a cohort of people with chronic HBV in a Vietnamese urban setting, as well as investigating potential associated factors, with the hope of identifying appropriate clinical and policy-level implications.

## 2. Materials and Methods 

### 2.1. Study Setting and Sampling Method

In this study, we collected data from a cross-sectional study, which was performed in October 2018 in the Chronic Hepatitis Clinic in the Viet-Tiep Hospital, Hai Phong, Vietnam. The inclusion criteria for selecting patients included being diagnosed with chronic hepatitis B (CHB), being aged 18 years old and above, agreeing to participate in the study and having the ability to communicate with the interviewers. Participants were excluded from the study if they suffered from severe health conditions which may have affected their ability to answer the questionnaire. The convenient sampling method was used to recruit patients, and a total of 298 participants agreed to be involved in the study.

### 2.2. Data Measurement

The 20 mininute face-to-face interviews were performed. Health staff in the clinic, who had undergone intensive training, were trained to interview the participants in order to secure the quality of the data. The confidentiality of the participants was ensured and written informed consent was obtained from the participants.

#### 2.2.1. Socioeconomic and Health Status 

Data on socioeconomic characteristics including gender, age, education, marital status, occupation, and monthly income were collected. Health status was self-reported based on the EuroQol -visual analog scale (EQ-VAS), rating from 0 (the worst health condition that you can imagine) to 100 (the best health condition that you can imagine).

#### 2.2.2. Substance Use

In order to assess alcohol use disorder, we used the Alcohol Use Disorders Identification Test-Consumption (AUDIT-C). Participants were asked about their frequency of alcohol drinking, standard alcohol drinking in a typical day, and the frequency of having six or more drinks with the total score ranging from 0 to 12. An AUDIT-C positive result was defined as male participants having a score of 4 or more and female participants having a score of 3 or more. Binge drinking was defined as participants having six or more drinks on one occasion. The current smoking status of patients was also explored by asking them to report their smoking status in the previous 30 days. The number of cigarettes per day was also examined. In this study, the information on cigarettes only focused on tobacco smoking.

#### 2.2.3. Sexual Practices

Participants were asked to report the number of sex partners in the last 12 months; the type of sex partners, including casual sexual partners (defined as sexual partners having sexual behavior outside of a committed/romantic relationship [17]), one-night stands (a single sexual encounter without further relations between the sexual participants [18]), and sex workers (people who receive money or goods in exchange for sexual services [19]); and whether they had used condoms in the last sexual intercourse within the last 12 months.

### 2.3. Statistical Analysis

Data analysis was conducted using STATA version 15.0 (Stata Corp. LP, College Station, TX, USA). Descriptive statistics were used to present the socioeconomic characteristics variables, alcohol and tobacco use, and the sexual practices among the participants. A chi-squared test was performed to compare gender differences. A multivariate logistic regression model was used to identify factors associated with risk behaviors, including monthly alcohol use or more, current smoker, and having multiple sex partners in the previous 12 months. The independent variables included socioeconomic characteristics (age, gender, marital status, educational level, occupation, and income level) and current smoking status. To explore a parsimonious regression model, we used stepwise backward selection strategies with the threshold of 0.2 for selecting variables. Statistical significance was acknowledged at a *p*-value of less than 0.05.

### 2.4. Ethical Approval

The protocol of this study was reviewed and approved by the Institutional Review Board of Hai Phong University of Medicine and Pharmacy.

## 3. Results

Table 1 illustrates the information regarding the socioeconomic characteristics and self-rated health status of participants. Half of the participants were male (54.5%), and 82.8% of participants were aged above 30. The majority of patients had a high school education and above (81%) and lived with a spouse/partner (89.2%). Of the participants, 36.6% were freelancers, which was followed by farmer/blue-collar workers and accounted for 34.1%. The mean EQ-VAS score was 74.5 (SD = 13.1).

Table 2 shows the alcohol and tobacco use among CHB patients. It shows that 82.5% of participants reported never using alcohol. Abstaining from alcohol was significantly and statistically higher among women than that of men (97% and 70.4%, respectively). The majority of patients drank from 0 to 2 cups of alcohol in a typical day (95.3%) and participants who drank 3–4 cups or 5–6 cups per day were only males (4.7%). Of the participants, 90.2% reported that they never had six or more drinks, and 11.7% of male participants drank six or more drinks less than monthly. No female patients had an AUDIT-C positive result while the percentage among male patients was 7.4%. Of the participants, 14.5% were current smokers, and the percentage of smoking among males (25.9%) was statistically and significantly higher than that of females (0.7%). The mean number of cigarettes per day was 7.4 (SD = 3.4).

The information regarding sexual practices among CHB patients is presented in Table 3. It shows that 77.2% of participants had sexual intercourse in the last 12 months. About half of the participants had one sex partner in the last 12 months, and 35.4% of male patients had sex with two or more sex partners. The percentage of male participants having sex with casual partners/one-night stands and sex workers was statistically and significantly higher than that of female patients. Approximately one-fifth of participants used condoms when having sex with their spouse/main partner (20.8%), and 66.7% and 74.1% of participants used condoms when having sex with casual partners/one-night stands and sex workers, respectively.

The reduced regression model is presented in Table 4. Female participants were less likely to use alcohol monthly (OR = 0.08; 95%CI = 0.02–0.25), be a current smoker (OR = 0.03; 95%CI = 0.00–0.22), or have multiple sex partners in the last 12 months (OR = 0.04; 95%CI = 0.01–0.13), compared to male participants. White-collar workers were also less likely to have more than one sex partner within the last 12 months (OR = 0.1; 95%CI = 0.01–0.88). Higher monthly individual income was associated with having multiple sex partners within the last 12 months (OR = 9.55; 95%CI = 2.66–34.35). Using alcohol monthly was positively related to being a current smoker (OR = 3.19; 95%CI = 1.3–7.83), and in contrast being a current smoker was also positively associated with monthly alcohol use or more (OR = 3.16; 95%CI = 1.3–7.68).

## 4. Discussion

The findings of this study contribute to the literature by adding information regarding smoking and drinking status and sexual practices among chronic hepatitis B patients. The participants who were female were less likely to have risky behavior in any dimension (monthly alcohol use, current smoker, or having multiple sex partners in the last 12 months). In terms of sexual practices, white-collar workers had a lower risk of having multiple sex partners in the last 12 months compared to participants who were unemployed, while people who had a higher individual income level were more likely to have multiple sex partners in the last 12 months. Monthly alcohol use was positively associated with being a current smoker, and in contrast, being a current smoker was also related to monthly alcohol use.

The percentage of current smokers in our study was relatively low and lower than the prevalence of smoking among Vietnamese males in the general population [20], as well as in previous studies which were conducted among the HBV population in China and Korea [21,22]. The disproportionate rates of cigarette smoking among CHB patients in other settings can be explained by the differences in socioeconomic status [23], the concurrence of other substance use disorders [13], as well as chronic smoking-related comorbidities [24] since the treatment duration of CHB patients is prolonged. This percentage was also lower than the proportion of smokers among HIV/AIDS patients in the previous study [25]. This higher proportion can be explained by the fact that among people infected with HIV/AIDS, especially those who are drug users, smoking and drug use are complementary by sharing similar cues and withdrawal symptoms [26]. Moreover, given the close relationship between smoking and chronic liver disease for instant hepatocellular carcinoma, smoking screening and support for smoking cessation should be integrated into the HBV treatment program in Vietnam. In another study among rural immigrants in Hanoi, Vietnam, there was a significantly higher likelihood of engaging in smoking and drinking behavior compared to our study [27].

In our study, percentage of people abusing alcohol was relatively lower than what has been reported in previous studies [28,29], which shows the result of the relationship between alcohol consumption and hepatitis B. In comparison with people infected with HIV/AIDS in Vietnam, our result also depicted a lower proportion of drinking alcohol than males receiving antiretroviral therapy treatment [30]. Alcohol is considered as the only cause for alcoholic liver disease, especially chronic viral hepatitis [31]. The progression of chronic liver disease to cirrhosis and hepatocellular carcinoma is significantly impacted by heavy alcohol consumption [13]. Moreover, alcohol use disorder may impair the response to medications when treating chronic hepatitis B [13]. Furthermore, smoking and being male were also considered as risk factors for drinking alcohol on a monthly basis, which is similar to a previous study [28]. People who smoke are more likely to drink alcohol, and by contrast, drinkers tend to smoke more heavily [32]. The co-use of alcohol and cigarette smoking increase the synergistic carcinogenic effects [33]. 

In this study, we highlighted that the percentage of participants having multiple sex partners was relatively low. However, the proportion of individuals using condoms when having sexual intercourse with casual sex partners or sex workers was low, which is consistent with a previous study of HIV/AIDS patients [34]. HBV can be easily transmitted by having sexual contact with an infected person, and it is also considered as one of the prevalent sexually transmitted infectious diseases, particularly among people who have multiple sex partners [35]. In our study, those working white-collar jobs were less likely to have multiple sex partners in the last 12 months, compared to those who were unemployed. Unemployment can be considered as a risk factor for multiple partners due to a deprived social background and excess free time, which in turn can induce more frequent casual sexual events [36,37]. 

Several public health suggestions can be drawn from this study. First, along with medical treatment care, more effort should be put in place to thoroughly address smoking and drinking among CHB patients. Since alcohol and smoking abstinence are recommended to slow the progression of chronic liver disease, counseling sessions should be promptly recommended and incorporated into treatment clinics. Education on safe sexual behavior, as well as sufficient distribution of condoms, should be integrated into healthcare services during hepatitis B treatment so that the risk behaviors and virus transmission can be reduced. In addition, safe-sex education should be focused on unemployed CHB patients, as they are more likely to have more than one sex partner.

There were several limitations to our study. Participants answered the questionnaire based on their recall ability regarding the number of cigarettes, alcohol consumption, and the number of sex partners in the last 12 months, which can cause recall bias. Secondly, the sampling method in this study was convenience sampling, which could possibly have decreased the generalizability of our findings. Thirdly, a cross-sectional study design cannot establish the causal relationship between risk factors and outcomes. Therefore, further research and longitudinal data are more appropriate to gain deeper knowledge and provide adequate explanations for the results. Several variables regarding risk factors should be included in future studies, such as injection drug users (IDU); data on the history of tobacco use disorder; type of intercourse—anal, vaginal, or oral; and orientation—heterosexual, bisexual, or homosexual.

## 5. Conclusions

This study highlights a low level of cigarette smoking and alcohol consumption but a high proportion of participants having sexual intercourse with casual sex partners or sex workers without using a condom. In order to decrease the risk of smoking and alcohol abuse among CHB patients, counseling sessions should be promptly recommended and incorporated into treatment clinics. To reduce virus transmission, education about safe sexual behaviors, along with sufficient condom distribution, should be integrated into healthcare services during CHB treatment.

## Figures and Tables

**Table 1 ijerph-16-00570-t001:** Socioeconomic characteristics and self-perceived health status among chronic hepatitis B (CHB) patients.

Characteristics	Male		Female		Total		*p*-Value
	*n*	%	*n*	%	*n*	%	
**Total**	162	54.5	135	45.5	**297**	**100.0**	
**Age group**							
≤30 years old	21	13	30	22.2	51	17.2	0.17
31–45 years old	37	22.8	32	23.7	69	23.2	
46–60 years old	52	32.1	38	28.1	90	30.3	
>60 years old	52	32.1	35	25.9	87	29.3	
**Education**							
<High school	22	13.7	34	25.4	56	19	<0.01
High school	82	50.9	43	32.1	125	42.4	
>High school	57	35.4	57	42.5	114	38.6	
**Marital status**							
Single	17	10.6	6	4.4	23	7.8	0.07
Having spouse/partner	141	87.6	123	91.1	264	89.2	
Divorce/Widow	3	1.9	6	4.4	9	3.0	
**Occupation**							
Unemployed	5	3.2	7	5.2	12	4.1	0.52
Freelancer	60	38.5	46	34.3	106	36.6	
White-collar workers	30	19.2	29	21.6	59	20.3	
Farmer/Blue-collar workers	51	32.7	48	35.8	99	34.1	
Others	10	6.4	4	3	14	4.8	
	**Mean**	**SD**	**Mean**	**SD**	**Mean**	**SD**	
**Age**	50.6	15.8	47.5	16.2	49.2	16.0	0.11
**Monthly personal income (mil VND)**	7.1	3.3	5.9	2.3	6.6	3.0	<0.01
**Self-rate health (0–100)**	73.5	14.1	75.6	11.8	74.5	13.1	0.34

**Table 2 ijerph-16-00570-t002:** Alcohol and tobacco use among CHB patients.

Characteristics	Male		Female		Total		*p*-Value
	*n*	%	*n*	%	*n*	%	
**Frequency of alcohol drinking**							
Never	114	70.4	131	97	245	82.5	<0.01
Monthly or less	29	17.9	4	3	33	11.1	
2–4 times a month	9	5.6	0	0	9	3	
2–3 times a week	6	3.7	0	0	6	2	
≥four times per week	4	2.5	0	0	4	1.3	
**Standard alcohol drinking in a typical day**							
0–2 cups	148	91.4	135	100	283	95.3	<0.01
3–4 cups	6	3.7	0	0	6	2	
5–6 cups	8	4.9	0	0	8	2.7	
**Frequency of having six or more drinks**							
Never	134	82.7	134	99.3	268	90.2	<0.01
Less than monthly	19	11.7	1	0.7	20	6.7	
Monthly	5	3.1	0	0	5	1.7	
Weekly	3	1.9	0	0	3	1	
Daily or almost daily	1	0.6	0	0	1	0.3	
**AUDIT-C positive**	12	7.4	0	0	12	4.0	**<0.01**
**Current smoking**	42	25.9	1	0.7	43	14.5	**<0.01**
	**Mean**	**SD**	**Mean**	**SD**	**Mean**	**SD**	
AUDIT-C score	0.9	1.7	0.0	0.2	0.5	1.4	<0.01
Number of cigarettes per day (*n* = 43)	7.4	3.4	6	0.0	7.4	3.4	0.76

**Table 3 ijerph-16-00570-t003:** Sexual practices among CHB patients.

Characteristics	Male		Female		Total		*p*-Value
	*n*	%	*n*	%	*n*	%	
**Having sexual intercourse in the last 12 months**	132	82	95	71.4	227	77.2	0.03
**Number of sex partners in the last 12 months**							
0	29	18.0	38	28.6	67	22.8	<0.01
1	75	46.6	90	67.7	165	56.1	
≥2	57	35.4	5	3.8	62	21.1	
**Types of sex partner in the last 12 months**							
Spouse/main partner (*n* = 227)	131	99.2	95	100	226	99.6	0.40
Casual partner/One-night stand (*n* = 227)	29	22.0	1	1.1	30	13.2	<0.01
Sex worker (*n* = 226)	27	20.5	0	0.0	27	11.9	<0.01
**Condom use in the last sexual intercourse within the last 12 months**							
Spouse/main partner (*n* = 226)	33	25.2	14	14.7	47	20.8	0.06
Casual partner/One-night stand (*n* = 30)	19	65.5	1	100	20	66.7	0.47
Sex worker (*n* = 27)	20	74.1	0	0	20	74.1	

**Table 4 ijerph-16-00570-t004:** Associated factors with risk behaviors among patients with CHB.

Characteristics	Monthly Alcohol Use or More		Current Smoker		Having Multiple Sex Partners in the Last 12 Months	
	OR	95%CI	OR	95%CI	OR	95%CI
**Age**	0.96 **	0.93; 1.00	0.99	0.96; 1.03	0.92 ***	0.89; 0.96
**Gender**						
Male						
Female	0.08 ***	0.02; 0.25	0.03 ***	0.00; 0.22	0.04 ***	0.01; 0.13
**Education**						
<High school						
High school	1.03	0.31; 3.47	0.68	0.21; 2.18	2.03	0.51; 8.13
>High school	1.81	0.46; 7.08	0.48	0.11; 2.03	1.45	0.30; 7.06
**Marital status**						
Single						
Having spouse/partner	0.79	0.22; 2.86	2.02	0.44; 9.36	0.43	0.09; 2.10
Divorce/Widow	0.90	0.04; 18.97	13.94 *	0.83; 234.49	1.15	0.05; 28.53
**Occupation**						
Unemployed						
Freelancer	0.67	0.05; 8.72	0.18	0.02; 1.94	0.18 *	0.03; 1.17
White-collar workers	0.35	0.02; 5.07	0.11 *	0.01; 1.39	0.10 **	0.01; 0.88
Farmer/Blue-collar workers	0.21	0.02; 2.85	0.09 *	0.01; 1.06	0.30	0.04; 2.00
Others	0.23	0.01; 7.66	0.07 *	0.00; 1.60	-	-
**Log monthly individual income**	2.10	0.71; 6.19	1.58	0.50; 4.95	9.55 ***	2.66; 34.35
**Monthly alcohol use or more**						
No						
Yes			3.19 **	1.30; 7.83	0.72	0.27; 1.93
**Current smoker**						
No						
Yes	3.16 **	1.30; 7.68				

*** *p* < 0.01, ** *p* < 0.05, * *p* < 0.1. OR: Odd ratio; 95%CI: Confidence Interval.

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
