# Peer review of "Understanding Risk Behaviors of Vietnamese Adults with Chronic Hepatitis B in an Urban Setting"

_ijerph, 2019, doi:10.3390/ijerph16040570_

Round 1

Reviewer 1 Report

This is a very important article in regard to risk factors for HBV and presents a firm basis for wanting to understand them as provided in the introduction. 

The overarching concern I have is the definition of major terms throughout. Some terms that need further defining include but are not limited to: "risky behaviors" "face to face interviews"

For substance use, why was AUDIT-C used? Why weren't NIAAA guidelines used to assess at risk drinking. AUDIT-C is used for identifying more at risk or potential alcohol use disorders.Why was "6 drinks" used as a cutoff in your interviewing?

For cigarette smoking: why was amount of cigarettes used in past 30 days used versus pack year history evaluated? This seems it would miss so many people who have history of tobacco use disorder. Also clarify is cigarette referring to tobacco, hashish, cannabis, etc?

Sexual practices needs much further clarification. Need to know type of intercourse, anal, vaginal, oral; orientation: heterosexual, bisexual, homosexual; these are very important risk factors. I don't feel that just examining number of partners is adequate. Also need to define "casual partner" "one night stand" "sex-worker" These terms are very ambiguous from an outsiders' point of view and must be clearly defined.

Questions I have are: why wasn't IVDU listed as a risk factor or asked of patients? Since there was mention of patients in methadone programs, IVDU as a risk factor needs to be assessed.

Discussion section: line 9 and 21, define substance and alcohol abuse respectively. Why was term "abuse" used? It is no longer in the DSM-5 criteria for substance use disorder diagnosis, this needs to be removed.

Very often throughout discussion it is stated "our previous studies showed." These studies need to be cited properly and explain as to why they are included here.

Grammatical/spelling changes: throughout the article numbers are written as ex} 122 thousand, this should be changed to 122,000. There are incorrect uses of prepositions and conjunctions throughout.

Line 56-58 does not make sense

Line 71-72 not a complete sentence

Line 81: "including" not used correctly

Author Response

Dear Reviewer, 

It is our great pleasure to receive your comments in a short period of time. Reviewer mentioned a number of interesting points that enabled us to work further on the revision of the manuscript. We have carefully discussed to address those issues in the following attached file. All authors have agreed with the alterations.

Reviewer 2 Report

Minor revisions are suggested.

In the Materials and Methods section, a more detailed description of the statistical analysis tools used in the study is recommended (page 3).

In the Discussion section, I would suggest some comparison of the results with the findings of previous studies concerning other Vietnamese populations (rural, immigrants). 

The syntax in the following sentences is not correct, impacting the comprehension of the meaning. Please modify: page 7, lines 2-5: "... regarding smoking and drinking status and sexual practices among chronic hepatitis B patients. Factors that positively associated with risk behaviors including gender, occupation, income level and the concurrence of smoking and drinking."

page 7, lines 12-13: "Because people infected [18] with HIV, especially who are drug users, smoking and drug use were complementary by sharing similar cues and withdrawal symptoms [22]."

In page 7, line 36, instead of "implications", please use a more appropriate word for the purposes of this study.

Author Response

(The authors gave the same response as above.)
